Prognostic factors and outcomes of invasive pulmonary aspergillosis, a retrospective hospital-based study

Chen Wei-Che 1
http://orcid.org/0000-0002-8345-5304 Chen I-Chieh 2
Chen Jun-Peng 2
Liao Tsai-Ling 2 3
http://orcid.org/0000-0001-7593-3065 Chen Yi-Ming 2 3 4 5 6 7 ymchen1@vghtc.gov.tw
1 Division of Nephrology, Department of Internal Medicine, Taichung Veterans General Hospital , Taiwan
2 Department of Medical Research, Taichung Veterans General Hospital , Taichung , Taiwan
3 Institute of Biomedical Science and Rong-Hsing Research Center for Translational Medicine, National Chung Hsing University, Taichung , Taichung , Taiwan
4 Division of Allergy, Immunology and Rheumatology, Taichung Veterans General Hospital , Taichung , Taiwan
5 Department of Post-Baccalaureate Medicine, National Chung Hsing University, Taichung , Taichung , Taiwan
6 Precision Medicine Research Center, National Chung Hsing University, Taichung , Taichung , Taiwan
7 School of Medicine, National Yang-Ming Chiao Tung University , Taipei , Taiwan
Nucera Francesco
Electronic publication date: 2024 Feb 28
Publication date: 2024
Volume: 12
Electronic Location ID: e17066
Received 2023 Oct 18; Accepted 2024 Feb 16
Copyright: © 2024 Chen et al.
Copyright year: 2024
Copyright holder: Chen et al.
License: This is an open access article distributed under the terms of the Creative Commons Attribution License, which permits unrestricted use, distribution, reproduction and adaptation in any medium and for any purpose provided that it is properly attributed. For attribution, the original author(s), title, publication source (PeerJ) and either DOI or URL of the article must be cited.
License URL: https://creativecommons.org/licenses/by/4.0/

Keywords: Invasive pulmonary aspergillosis, Respiratory failure, Renal failure, Mortality, Risk factors

Funding: National Science and Technology Council, Taiwan MOST 110-2314-B-075A-010, 110-2634-F-A49-005 and NSTC 111-2314-B-005-007-MY3 Academia Sinica, Taiwan VTA111-V2-1-1 Taichung Veterans General Hospital, Taiwan TCVGH-1127301C, TCVGH-1127302D, TCVGH-YM1120110, TCVGH-1127304B and VTA113-V2-1-1 This study was funded by the National Science and Technology Council, Taiwan (MOST 110-2314-B-075A-010, 110-2634-F-A49-005, NSTC 111-2314-B-005-007-MY3), the Academia Sinica, Taiwan (VTA111-V2-1-1), and the Taichung Veterans General Hospital, Taiwan TCVGH-1127301C, TCVGH-1127302D, TCVGH-YM1120110, TCVGH-1127304B and VTA113-V2-1-1. The funders had no role in study design, data collection and analysis, decision to publish, or preparation of the manuscript.

==============================
Objective

Invasive pulmonary aspergillosis (IPA) affects immunocompromised hosts and is associated with higher risks of respiratory failure and mortality. However, the clinical outcomes of different IPA types have not been identified.

Methods

Between September 2002 and May 2021, we retrospectively enrolled patients with IPA in Taichung Veterans General Hospital, Taiwan. Cases were classified as possible IPA, probable IPA, proven IPA, and putative IPA according to EORTC/MSGERC criteria and the AspICU algorithm. Risk factors of respiratory failure, kidney failure, and mortality were analyzed by logistic regression. A total of 3-year survival was assessed by the Kaplan-Meier method with log-rank test for post-hoc comparisons.

Results

We included 125 IPA patients (50: possible IPA, 47: probable IPA, 11: proven IPA, and 17: putative IPA). Comorbidities of liver cirrhosis and solid organ malignancy were risk factors for respiratory failure; diabetes mellitus and post-liver or kidney transplantation were related to kidney failure. Higher galactomannan (GM) test optical density index (ODI) in either serum or bronchoalveolar lavage fluid was associated with dismal outcomes. Probable IPA and putative IPA had lower 3-year respiratory failure-free survival compared to possible IPA. Probable IPA and putative IPA exhibited lower 3-year renal failure-free survival in comparison to possible IPA and proven IPA. Putative IPA had the lowest 3-year overall survival rates among the four IPA groups.

Conclusion

Patients with putative IPA had higher mortality rates than the possible, probable, or proven IPA groups. Therefore, a prompt diagnosis and timely treatment are warranted for patients with putative IPA.

Introduction

Invasive pulmonary aspergillosis (IPA) is a major respiratory infection affecting patients with hematologic diseases, hematopoietic stem cell transplantation, prolonged use of corticosteroids, and solid organ transplantation (Donnelly et al., 2020). The estimated global burden of IPA was about 250,000 cases (Bongomin et al., 2017) and the economic load was considerable, ranging from 25,929 to 100,730 US dollars per patient hospitalization in patients with invasive aspergillosis (Kim, Nicolau & Kuti, 2011). Despite improvements in diagnostic tools, early identification of IPA is still challenging, and delayed treatment may be associated with a devastating outcome. The reported mortality rates of IPA range from 44% to 80% (Meersseman et al., 2004; Schauwvlieghe et al., 2018; Subirà et al., 2002; Bartoletti et al., 2021). However, the prevalence of IPA could be underestimated due to lack of clinical suspicion, variability in diagnostic testing, and the concomitant use of immunosuppressive agents. Prognostic factors associated with mortality included advanced age, higher C-reactive protein (CRP) value, acute kidney failure, and acute respiratory failure (Tong et al., 2021; Corcione et al., 2021). Other associated comorbidities of chronic kidney disease (CKD), diabetes mellitus (DM), and connective tissue disease were also related to worse outcomes (Silva et al., 2012). Aside from the traditional risk factors, patients with chronic obstructive pulmonary disease (COPD), acute liver failure, and liver cirrhosis were at risk of IPA (Arai et al., 2014; Verma et al., 2019; Yang et al., 2021; Barberan et al., 2012; Blot et al., 2012).

Consensus definitions of IPA were based on the European Organization for Research and Treatment of Cancer/Mycosis Study Group Education and Research Consortium (EORTC/MSGERC) and the AspICU algorithm (Donnelly et al., 2020; Blot et al., 2012). IPA were categorized as proven, probable, and possible IPA by the EORTC/MSGERC criteria or putative IPA by the AspICU criteria. These criteria included host factors, symptoms or radiological features, and mycological evidence. Clinical manifestations and sometimes the radiological presentation can be non-specific (Huang et al., 2018). Furthermore, obtaining a CT scan is difficult in patients with unstable hemodynamic status, which makes a correct diagnosis even more challenging. The serum galactomannan antigen (GM) test is valuable for patients with neutropenic conditions, those undergoing bone marrow transplantation, or individuals with cancer (Mercier et al., 2020; Bassetti, Peghin & Vena, 2018). Yet, it is essential to recognize that its sensitivity is significantly compromised in the context of azole prophylactic therapy (Mercier et al., 2020).

A prior study in a case series of pulmonary aspergillosis showed that the prevalence of mechanical ventilation was approximately 25% (Iqbal et al., 2016), whereas, in patients with IPA, the rates of intubation could be up to 50% (Taccone et al., 2015). Previous studies demonstrated that IPA patients needing mechanical ventilation support or renal replacement therapy had higher mortality rates (Taccone et al., 2015; Soontrapa, Chongtrakool & Chayakulkeeree, 2022). In a multicenter study, the prevalence of renal replacement therapy in probable IPA or proven IPA was around 27% (Pardo et al., 2019). Notably, this figure may be subject to bias due to the specific focus on individuals within critical care settings. These studies emphasized the outcomes of probable IPA, proven IPA, and putative IPA. However, whether differential clinical outcomes exist among patients categorized by EORTC/MSGERC or AspICU criteria remains unknown. Moreover, factors relevant to mechanical ventilation and hemodialysis have yet to be elucidated. No prior study had conducted comparisons of clinical outcomes of these four disease categories. In addition, the endpoint of respiratory failure or renal failure remains unknown for these four pulmonary aspergillosis classifications.

Our study purpose was to identified risks associated dismal outcomes in patients with IPA. We aimed to evaluate the prognostic factors for respiratory failure, kidney failure, and patient survival of different diagnostic criteria (possible IPA, probable IPA, proven IPA, and putative IPA) in a hospital-based population.

Materials and Methods

Study population

We conducted a retrospective study of patients diagnosed with IPA in Taichung Veterans General Hospital, Taiwan from September 2002 to May 2021. Among these patients, 50 were diagnosed as possible IPA by the EORTC/MSGERC criteria (Donnelly et al., 2020) using the following conditions: 1) Presence of one of the following host factors: hematologic malignancy, post-allogeneic stem cell transplant, post-solid organ transplant, recent history of neutropenia <500 neutrophils/mm3 for >10 days, prolonged use of corticosteroids at a dose of ≥0.3 mg/kg for ≥3 weeks in the past 60 days, treatment with recognized T-cell immunosuppressants during the past 90 days, treatment with B-cell-depleting immunosuppressants, acute graft-versus-host disease grade III or IV involving the gut, lungs, or liver that is refractory to first-line treatment with steroids, inherited severe immunodeficiency.

2) Presence of one of the following clinical features: patterns on CT, including dense, well-circumscribed lesion(s) with or without a halo sign, cavity, air crescent sign, wedge-shaped and segmental or lobar consolidation.

Forty-seven patients had a diagnosis of probable IPA by meeting the conditions below: 1) Presence of host factors as described above.

2) Presence of clinical features as described above.

3) Mycological evidence (any Aspergillus species revealed by culture from sputum, bronchial brush, bronchial alveolar lavage (BAL), or aspirate; galactomannan antigen optical density index (ODI) value of ≥1 in serum, ODI ≥ 1 in BAL fluid, or ODI ≥ 0.7 in serum with BAL fluid ≥0.8; 2 or more positive Aspergillus PCR tests or one positive Aspergillus PCR test in serum with one positive Aspergillus PCR test in BAL fluid).

Eleven were diagnosed as proven IPA by the following conditions: 1) Specimen from sterile lung showed positive culture, PCR, or DNA sequencing for Aspergillus species.

2) Cytopathologic, histopathologic, or direct microscopic examination of a specimen acquired by needle aspiration or biopsy in which hyphae are seen with associated tissue damage.

Seventeen did not meet the EORTC/MSGERC criteria but were deemed to be putative IPA based on the AspICU algorithm (Blot et al., 2012) under the following conditions: 1) Lower respiratory tract specimen culture yields Aspergillus species.

2) Compatible signs and symptoms (dyspnea, hemoptysis, fever refractory to at least 3 days of appropriate antibiotic therapy, pleuritic chest pain, pleuritic rub, and worsening respiratory insufficiency in spite of antibiotic therapy and ventilator support).

3) Abnormal chest X-ray or CT scan of the lungs.

4) Either one of the host risk factors (neutropenia, underlying hematological or oncological malignancy treated with cytotoxic agents, congenital or acquired immunodeficiency, receiving glucocorticoid treatment with prednisolone equivalent >20 mg/day), or positive mycological test (Aspergillus-positive culture of BAL fluid with positive cytological smear showing branching hyphae).

This retrospective study was approved by the Ethics Committee of Clinical Research, Taichung Veterans General Hospital (CE21478A). The study was carried out in accordance with the Declaration of Helsinki.

Data collection

Data were extracted from the electronic health records of Taichung Veterans General Hospital. Patient data included demographics, laboratory profile, comorbidities, and outcomes. The index date was defined as the diagnostic date of IPA. The laboratory data comprised white blood cell count (WBC), hemoglobin, platelet, CRP, albumin, creatinine, total bilirubin, and galactomannan antigen with either serum or bronchoalveolar lavage fluid sample, which were collected within 6 months before the index date. Comorbidities, including asthma (ICD-9: 493.x; ICD-10: J45.x), COPD (ICD-9: 496; ICD-10: J44.x), DM (ICD-9: 250.x; ICD-10: E08-E13), solid organ malignancy (ICD-9: 140.x – 208.x; ICD-10: C00-D49), liver cirrhosis (ICD-9: 571; ICD-10:K74.6x), and hematological disease (ICD-9: 200.x – 209.x; ICD-10: C81.x – C96.x) were determined using ICD-9 /ICD-10 codes during hospitalization within 6 months before the index date.

Galactomannan test

Galactomannan antigen test was performed by enzyme-linked immunosorbent assay (Bio-Rad, Marnes-la-Coquette, France). The positivity of the GM test was interpreted according to international consensus criteria (Donnelly et al., 2020). The first GM test was collected within 2 weeks of the index date. The maximal GM test was defined as the highest value among serial GM tests performed 2 weeks before or 6 months after the index date.

Outcome determination

The study outcomes included the occurrence of respiratory failure, kidney failure, and mortality. We defined respiratory failure as the requirement for mechanical ventilation using the procedure code of ventilator; renal failure as the requirement of renal replacement therapy, including procedure codes for hemodialysis, peritoneal dialysis, kidney transplant, and ICD codes for uremia and end-stage renal disease (ICD-9: 585, 586, ICD-10: N18.X, N19.X) after the index date. Three-year survival rates were obtained from an electronic database maintained by Taiwan’s Ministry of Health and Welfare.

Statistical analysis

Descriptive statistics were used for all study variables. Discrete variables were expressed as number (percentage) and continuous variables as the median (interquartile range). The selected parameters were compared among the possible IPA, probable IPA, proven IPA, and putative IPA groups using the Chi-square test or Kruskal-Wallis test. Logistic regression analysis with adjustments for age and sex was used to determine the hazard ratio (HR) and 95% confidence interval (CI) for respiratory failure, renal failure, and three-year mortality. Three-year survival of the four populations were analyzed by Kaplan-Meier method with the log-rank test for post-hoc comparisons. Two-sided p values < 0.05 were considered statistically significant. Statistical analyses were performed using the Statistical Package for the Social Sciences, version 22.0 (SPSS; IBM Corp., Armonk, NY, USA) and MedCalc® Statistical Software version 20.014 (MedCalc Software Ltd, Ostend, Belgium).

Results

Comparisons of demographics, comorbidities, and outcomes among the IPA groups

A total of 125 patients with IPA were enrolled in this study. The demographic characteristics are shown in Table 1. The average follow-up period was 2.0 ± 2.8 years. We found that serum creatinine (p = 0.028), WBC (p = 0.001), and ODI of the GM test in serum (p < 0.001) or BAL fluid (p < 0.001) samples were the highest in the putative IPA group compared with the other three groups. The highest hemoglobin value (p < 0.001) was observed in the proven IPA group. The highest prevalence of hematologic disease (p < 0.001) was found in the possible IPA group compared with the other groups. Age, CRP, platelet, total bilirubin, and the prevalence of underlying COPD, asthma, liver cirrhosis, DM, and solid organ malignancy among the four IPA groups were similar.

Table 1 Demographic data of patients with possible IPA, probable IPA, proven IPA, and putative IPA.

	Possible IPA (n = 50)	Probable IPA (n = 47)	Proven IPA (n = 11)	Putative IPA (n = 17)	p value	
Gender (n = 125)									0.073	
Female	25	(50.0%)	26	(55.3%)	3	(27.3%)	4	(23.5%)		
Male	25	(50.0%)	21	(44.7%)	8	(72.7%)	13	(76.5%)		
Age, years (n = 125)	61.0	(41.0–68.3)	54.0	(43.0–69.0)	59.0	(54.0–77.0)	74.0	(46.0–77.5)	0.142	
Lab data										
Creatinine (n = 115)	0.8	(0.7–1.1)	0.9	(0.7–1.6)	0.8	(0.6–1.0)	1.2	(0.9–1.5)	0.028*	
Hemoglobin (n = 125)	9.7	(7.9–10.8)	9.7	(8.1–11.5)	13.2	(11.4–14.4)	10.1	(8.9–11.4)	<0.001**	
CRP (n = 102)	5.2	(1.8–12.4)	4.0	(0.6–9.8)	2.3	(0.2–24.0)	16.7	(0.8–27.1)	0.193	
Platelet (n = 125)	115.0	(45.0–239.3)	143.0	(61.0–237.0)	188.0	(126.0–372.0)	159.0	(80.0–289.5)	0.120	
WBC (n = 125)	4,455.0	(2,665.0–8,465.0)	6,330.0	(3,780.0–9,850.0)	10,400.0	(4,040.0–15,400.0)	13,960.0	(8,055.0–20,075.0)	0.001**	
Total bilirubin (n = 124)	0.4	(0.4–0.8)	0.6	(0.4–0.9)	0.5	(0.3–0.9)	0.6	(0.4–1.2)	0.451	
Albumin (n = 85)	3.9	(3.0–4.3)	3.5	(2.9–4.0)	3.8	(2.9–4.2)	3.8	(3.0–4.2)	0.635	
Galactomannan test (n = 110)	
Blood-First	0.08	(0.06–0.17)	0.29	(0.11–0.66)	0.42	(0.16–1.04)	0.26	(0.09–6.18)	<0.001**	
Blood-Max	0.12	(0.08–0.27)	1.04	(0.51–2.73)	0.47	(0.16–1.22)	4.16	(0.42–7.40)	<0.001**	
BAL-First	0.11	(0.07–0.33)	0.23	(0.08–2.23)	0.78	(0.08–1.00)	6.44	(1.19–8.78)	<0.001**	
BAL-Max	0.16	(0.07–0.39)	0.78	(0.10–2.93)	0.78	(0.25–2.81)	7.02	(2.13–8.78)	<0.001**	
Comorbidity (n = 125)										
COPD	4	(8.0%)	4	(8.5%)	2	(18.2%)	4	(23.5%)	0.262	
Asthma	3	(6.0%)	8	(17.0%)	1	(9.1%)	2	(11.8%)	0.389	
Liver cirrhosis	1	(2.0%)	3	(6.4%)	0	(0.0%)	0	(0.0%)	0.441	
DM	9	(18.0%)	11	(23.4%)	1	(9.1%)	2	(11.8%)	0.589	
Hematological disease	39	(78.0%)	24	(51.1%)	2	(18.2%)	1	(5.9%)	<0.001**	
Solid organ malignancy	7	(14.0%)	10	(21.3%)	3	(27.3%)	5	(29.4%)	0.479	
Liver or kidney transplant	2	(4.0%)	4	(8.5%)	0	(0.0%)	0	(0.0%)	0.408	
Notes:

Chi-square test or Kruskal-Wallis test. Median (IQR).

* p < 0.05.

** p < 0.01.

IPA, invasive pulmonary aspergillosis; CRP, C-reactive protein; WBC, white blood cell; BAL, bronchial alveolar lavage; COPD, chronic obstructive pulmonary disease; DM, diabetes mellitus.

Risks for respiratory failure

Risk factors for mechanical ventilation are shown in Table 2. After adjustment for sex and age, higher first GM test ODI in BAL fluid samples (HR 1.17, 95% CI [1.00–1.37], p < 0.05), liver cirrhosis (HR 4.04, 95% CI [1.14–14.29], p < 0.05), and solid organ malignancy (HR 5.22, 95% CI [1.98–13.75], p < 0.01) were independent risks for respiratory failure. Patients classified as probable IPA (HR 3.68, 95% CI [1.18–11.45], p < 0.05) had higher risk for respiratory failure compared to those classified as possible IPA.

Table 2 Risk factors of respiratory failure.

	Univariate	Age- & sex-adjusted	
HR	95% CI	p value	HR	95% CI	p value	
Lab data							
Creatinine	1.12	[0.86–1.47]	0.406	1.12	[0.86–1.46]	0.407	
Hemoglobin	0.94	[0.79–1.13]	0.529	0.91	[0.76–1.11]	0.354	
CRP	0.91	[0.82–1.01]	0.089	0.90	[0.81–1.01]	0.082	
Platelet	1.00	[1.00–1.00]	0.890	1.00	[1.00–1.00]	0.805	
WBC	1.00	[1.00–1.00]	0.161	1.00	[1.00–1.00]	0.110	
Total bilirubin	1.13	[1.01–1.26]	0.026*	1.12	[1.00–1.25]	0.052	
Albumin	0.82	[0.42–1.60]	0.566	0.75	[0.37–1.52]	0.422	
Galactomannan test	
Blood-first	0.97	[0.72–1.30]	0.831	0.99	[0.74–1.33]	0.969	
Blood-max	1.08	[0.92–1.27]	0.340	1.08	[0.92–1.27]	0.353	
BAL-first	1.16	[0.99–1.35]	0.066	1.17	[1.00–1.37]	0.044*	
BAL-max	1.11	[0.95–1.31]	0.192	1.13	[0.96–1.34]	0.139	
Comorbidity							
COPD	1.23	[0.36–4.17]	0.739	0.94	[0.26–3.45]	0.931	
Asthma	0.29	[0.04–2.16]	0.226	0.32	[0.04–2.39]	0.266	
Liver cirrhosis	4.85	[1.43–16.41]	0.011*	4.04	[1.14–14.29]	0.030*	
DM	0.64	[0.19–2.17]	0.474	0.60	[0.17–2.04]	0.410	
Hematological disease	0.44	[0.19–1.06]	0.066	0.49	[0.20–1.21]	0.120	
Solid organ malignancy	4.81	[2.02–11.49]	<0.001**	5.22	[1.98–13.75]	0.001**	
Liver or kidney transplant	1.70	[0.40–7.31]	0.473	1.25	[0.28–5.58]	0.774	
IPA groups							
Possible IPA	Reference	Reference	
Probable IPA	3.53	[1.14–10.98]	0.029*	3.68	[1.18–11.45]	0.024*	
Proven IPA	1.99	[0.36–10.87]	0.428	1.68	[0.30–9.45]	0.556	
Putative IPA	4.36	[1.08–17.60]	0.039*	4.09	[0.98–17.03]	0.053	
Notes:

Cox proportional hazard regression.

* p < 0.05.

** p < 0.01.

CRP, C-reactive protein; WBC, white blood cell; BAL, bronchial alveolar lavage; COPD, chronic obstructive pulmonary disease; DM, diabetes mellitus; IPA, invasive pulmonary aspergillosis.

Figure 1A depicts the 3-year respiratory failure-free survival rates: 91.2% in patients with possible IPA, 71.7% in those with probable IPA, 90.0% in proven IPA, and 76.0% in putative IPA. Probable IPA and putative IPA had lower 3-year respiratory failure-free survival rates compared to possible IPA (p = 0.03 by Kaplan-Meier analysis).

Figure 1 Comparisons of respiratory failure, renal failure and overall survival among patients with possible, probable, proven, and putative IPA.

Comparisons of 3-year (A) respiratory failure-free survival, (B) renal failure-free survival, (C) overall survival among patients with possible, probable, proven, and putative IPA by Kaplan-Meier survival analysis. Post-hoc pairwise comparisons by log-rank test: (A) probable IPA vs. possible IPA, p = 0.009; putative IPA vs. possible IPA, p = 0.014 (B) probable IPA vs. possible IPA, p = 0.019; putative IPA vs. possible IPA, p = 0.001; probable IPA vs. proven IPA, p = 0.042; putative IPA vs. proven IPA, p = 0.026 (C) putative IPA vs. possible IPA, p = 0.005; putative IPA vs. probable IPA, p = 0.005; putative IPA vs. proven IPA, p = 0.006.

Risks for kidney failure

Table 3 shows that higher serum creatinine value (HR: 1.45, 95% CI [1.27–1.66], p < 0.01), higher first GM test ODI (HR 1.27, 95% CI [1.11–1.46], p < 0.01; HR 1.18, 95% CI [1.03–1.35], p < 0.05, respectively), higher maximal GM test ODI (HR 1.13, 95% CI [1.01–1.27], p < 0.05; HR 1.17, 95% CI [1.02–1.34], p < 0.05, respectively) in either serum or BAL fluid samples, DM (HR 2.07, 95% CI [1.12–3.82], p < 0.05), and post-liver or kidney transplantation (HR 3.45, 95% CI [1.37–8.65], p < 0.01) were independent risks for renal replacement therapy. Moreover, higher hemoglobin levels (HR 0.87, 95% CI [0.77–0.99], p < 0.05) and hematological disease (HR 0.44, 95% CI [0.25–0.79], p < 0.01) were associated with lower risk for renal failure. Patients classified as probable IPA and putative IPA had higher risk for renal failure compared to those classified as possible IPA (HR 2.17, 95% CI [1.12–4.20], p < 0.05; HR 2.17, 95% CI [1.12–4.20], p < 0.01, respectively).

Table 3 Risk factors of kidney failure.

	Univariate	Age- & sex-adjusted	
HR	95% CI	p value	HR	95% CI	p value	
Lab data							
Creatinine	1.43	[1.25–1.62]	<0.001**	1.45	[1.27–1.66]	<0.001**	
Hemoglobin	0.91	[0.81–1.02]	0.116	0.87	[0.77–0.99]	0.039*	
CRP	0.98	[0.94–1.02]	0.237	0.98	[0.94–1.01]	0.208	
Platelet	1.00	[1.00–1.00]	0.154	1.00	[1.00–1.00]	0.124	
WBC	1.00	[1.00–1.00]	0.893	1.00	[1.00–1.00]	0.856	
Total bilirubin	1.05	[0.96–1.16]	0.298	1.07	[0.97–1.18]	0.172	
Albumin	0.71	[0.46–1.10]	0.128	0.72	[0.46–1.13]	0.151	
Galactomannan test	
Blood-first	1.29	[1.13–1.48]	<0.001**	1.27	[1.11–1.46]	0.001**	
Blood-max	1.14	[1.02–1.28]	0.026*	1.13	[1.01–1.27]	0.037*	
BAL-first	1.17	[1.02–1.33]	0.025*	1.18	[1.03–1.35]	0.018*	
BAL-max	1.16	[1.02–1.33]	0.028*	1.17	[1.02–1.34]	0.021*	
Comorbidity							
COPD	1.13	[0.48–2.64]	0.786	0.99	[0.40–2.46]	0.980	
Asthma	1.40	[0.66–2.99]	0.380	1.36	[0.63–2.93]	0.438	
Liver cirrhosis	2.51	[0.78–8.10]	0.125	2.67	[0.81–8.82]	0.107	
DM	2.24	[1.23–4.06]	0.008**	2.07	[1.12–3.82]	0.021*	
Hematological disease	0.43	[0.25–0.77]	0.004**	0.44	[0.25–0.79]	0.006**	
Solid organ malignancy	0.91	[0.44–1.87]	0.792	0.82	[0.39–1.72]	0.597	
Liver or kidney transplant	3.07	[1.30–7.27]	0.011*	3.45	[1.37–8.65]	0.008**	
IPA groups							
Possible IPA	Reference	Reference	
Probable IPA	2.18	[1.13–4.21]	0.020*	2.17	[1.12–4.20]	0.021*	
Proven IPA	0.51	[0.11–2.23]	0.368	0.45	[0.10–2.01]	0.295	
Putative IPA	3.96	[1.73–9.05]	0.001**	3.41	[1.45–8.01]	0.005**	
Notes:

Cox proportional hazard regression.

* p < 0.05.

** p < 0.01.

CRP, C-reactive protein; WBC, white blood cell; BAL, bronchial alveolar lavage; COPD, chronic obstructive pulmonary disease; DM, diabetes mellitus; IPA, invasive pulmonary aspergillosis.

Figure 1B shows the 3-year renal failure-free survival rates: 66.8% in patients with possible IPA, 34.9% in those with probable IPA, 80.8% in proven IPA, and 37.6% in putative IPA. Probable IPA and putative IPA had lower 3-year renal failure-free survival rates compared to possible IPA or proven IPA (p = 0.001 by Kaplan-Meier analysis).

Risks for mortality

Table 4 shows that higher CRP levels (HR 1.03, 95% CI [1.01–1.06], p < 0.05), higher first GM test ODI in BAL fluid samples (HR 1.25, 95% CI [1.09–1.44], p < 0.01), and higher maximal GM test ODI in serum or BAL fluid samples (HR 1.15, 95% CI [1.04–1.27], p < 0.01; HR 1.27, 95% CI [1.11–1.46], p < 0.01, respectively) were independent risks for mortality. Putative IPA (HR 2.58, 95% CI [1.28–5.23], p < 0.01) had higher risk for mortality compared to possible IPA.

Table 4 Risk factors of mortality.

	Univariate	Age- & sex-adjusted	
HR	95% CI	p value	HR	95% CI	p value	
Lab data							
Creatinine	1.09	[0.91–1.30]	0.340	1.07	[0.89–1.29]	0.467	
Hemoglobin	0.96	[0.87–1.06]	0.439	0.92	[0.83–1.03]	0.137	
CRP	1.03	[1.01–1.06]	0.012*	1.03	[1.01–1.06]	0.013*	
Platelet	1.00	[1.00–1.00]	0.253	1.00	[1.00–1.00]	0.184	
WBC	1.00	[1.00–1.00]	0.041*	1.00	[1.00–1.00]	0.056	
Total bilirubin	0.98	[0.85–1.13]	0.809	0.98	[0.85–1.13]	0.762	
Albumin	0.78	[0.51–1.20]	0.260	0.80	[0.50–1.26]	0.334	
Galactomannan test	
Blood-First	1.05	[0.90–1.23]	0.506	1.08	[0.91–1.27]	0.371	
Blood-Max	1.16	[1.05–1.27]	0.003**	1.15	[1.04–1.27]	0.005**	
BAL-First	1.17	[1.04–1.31]	0.011*	1.25	[1.09–1.44]	0.002**	
BAL-Max	1.19	[1.06–1.33]	0.003**	1.27	[1.11–1.46]	0.001**	
Comorbidity							
COPD	1.24	[0.64–2.42]	0.529	0.97	[0.48–1.96]	0.930	
Asthma	0.51	[0.22–1.19]	0.119	0.54	[0.23–1.26]	0.154	
Liver cirrhosis	0.78	[0.19–3.19]	0.731	0.71	[0.17–2.93]	0.633	
DM	0.75	[0.40–1.40]	0.363	0.67	[0.36–1.25]	0.210	
Hematological disease	0.90	[0.56–1.43]	0.642	1.03	[0.63–1.67]	0.905	
Solid organ malignancy	1.38	[0.78–2.46]	0.269	1.18	[0.65–2.13]	0.593	
Liver or kidney transplant	0.37	[0.09–1.51]	0.166	0.27	[0.07–1.14]	0.074	
IPA groups							
Possible IPA	Reference	Reference	
Probable IPA	1.05	[0.62–1.78]	0.850	1.07	[0.63–1.81]	0.801	
Proven IPA	0.52	[0.18–1.48]	0.217	0.42	[0.15–1.23]	0.114	
Putative IPA	2.68	[1.35–5.30]	0.005**	2.58	[1.28–5.23]	0.008**	
Notes:

Cox proportional hazard regression.

* p < 0.05.

** p < 0.01.

CRP, C-reactive protein; WBC, white blood cell; BAL, bronchial alveolar lavage; COPD, chronic obstructive pulmonary disease; DM, diabetes mellitus; IPA, invasive pulmonary aspergillosis.

Figure 1C shows the 3-year overall survival rates: 40.4% in patients with possible IPA, 39.0% in those with probable IPA, 58.9% in proven IPA, and 20.2% in putative IPA. Putative IPA had lower 3-year overall survival rates compared to the other three IPA groups (p = 0.004 by Kaplan-Meier analysis).

Discussion

Our study identified crucial prognostic factors associated with IPA. We also demonstrated a differential risk of respiratory failure, kidney failure, and mortality among four groups of IPA. Patients with putative IPA exhibited the worst outcome in terms of mortality, followed by probable IPA. Moreover, patients with possible and proven IPA had superior outcomes compared with their counterparts. Our results provided insights for clinicians to identify potential high-risk patients and may help to guide timely anti-fungal therapeutics.

Few studies have investigated outcomes among the four IPA groups. In a cohort study, patients with putative IPA or proven IPA were found to have similar outcomes; both had higher mortality when compared to Aspergillus colonization (Soontrapa, Chongtrakool & Chayakulkeeree, 2022). In a randomized trial evaluating liposomal amphotericin B efficacy against invasive fungal disease (IFD) using EORTC/MSG 2008 definitions, a superior survival was observed in possible IFD than in probable or proven IFD (Cornely et al., 2011). Consistent with the two aforementioned studies (Soontrapa, Chongtrakool & Chayakulkeeree, 2022; Cornely et al., 2011), our results revealed that putative IPA had the worst outcome, even in the absence of traditional host immunocompromised factors, i.e., organ transplantation or hematological conditions. In addition, we provided outcomes of respiratory failure-free and renal failure-free survival: possible IPA had better respiratory failure-free survival rates and renal failure-free survival rates than probable IPA or putative IPA, but had similar outcomes in comparison with proven IPA. Moreover, the present study is the first investigation to compare clinical outcomes among four groups of IPA, and our findings provide crucial information, which may be useful in the management of Aspergillus diseases.

We found a 3-year overall survival rate of 39% in patients with probable IPA, 58.9% in those with proven IPA, and 20.2% in putative IPA. In a study of putative IPA, the 21-day mortality rate was 51.3% (Corcione et al., 2021). In another study, the mortality rate in critically ill probable IPA patients with underlying liver cirrhosis was 100% within the 2-year observation period (Lahmer et al., 2019). Pardo et al. (2019) reported 90-day survival rates of 23.7% in patients with probable IPA or proven IPA. In a single-institution retrospective study, the all-cause mortality rate was 27.4% in patients with proven IPA over the 7-year follow-up period (Tong et al., 2021). The present study provides valuable data of long-term outcomes in patients with IPA and multiple comorbidities. We believe that the different comorbidities of the enrolled participants and the availability of effective anti-fungal agents may affect the mortality rates of patients with IPA.

Comorbidities of DM, human immunodeficiency virus infection, CKD, COPD, hematological disease, and solid organ transplant could be associated with IPA (Iqbal et al., 2016; Taccone et al., 2015; Soontrapa, Chongtrakool & Chayakulkeeree, 2022; Pardo et al., 2019; Zhang et al., 2018). Prolonged neutropenia and steroid use were also risk factors of IPA (Donnelly et al., 2020). COPD patients were especially at risk for putative IPA because of the disease characteristics of chronic inflammatory, lung parenchymal destruction, and alteration of the microbiome (Ader, 2010; Kolwijck & van de Veerdonk, 2014). Lymphoproliferative disorder has been identified as a risk factor for mortality in patients with IPA (Taccone et al., 2015). Age >60 years, presence of pleural effusion, chemotherapy, and CRP > 14.1 mg/dL were found to be independent risk factors for mortality in proven IPA patients (Tong et al., 2021). Our results demonstrated similar comorbidities in IPA patients as those reported in previous studies. Furthermore, we found that liver cirrhosis and solid organ malignancy were correlated with respiratory failure, and DM and post-liver or kidney transplantation were associated with kidney failure in patients with IPA. Higher CRP levels were associated with mortality in IPA (Tong et al., 2021), which was consistent with our result. Surprisingly, our data demonstrated that hematological disease was a protective factor for renal failure. This observation could be attributed to the prompt diagnosis and timely initiation of treatment within these specific patient populations. In future studies, a larger sample size is needed to confirm our findings. Additionally, the increased WBC and creatinine levels detected in the putative IPA group might be linked to the partial exclusion of neutropenic patients and the critical illness status prevalent among most individuals in this specific group.

Patients with IPA had higher rates of GM test positivity in serum samples compared to those without IPA (Taccone et al., 2015). Serum GM kinetics correlated with treatment outcomes in hematology patients with invasive aspergillosis (Mercier et al., 2020). Post-renal transplant patients with IPA had lower survival rates when serum GM test ODI ≥ 2 or BAL GM test ODI ≥ 5 (Seok et al., 2020). Our results provide evidence that the first or maximal GM test ODI in either blood or BAL fluid are prognostic. We also found that a higher GM test ODI, except the first GM test ODI of blood, was an independent risk for mortality. Moreover, a higher first GM test ODI in BAL fluid was related to mechanical ventilation; higher first or maximal GM test ODI in either blood or BAL fluid was associated with kidney failure. Taken together, our data provide a comprehensive overview of the differential prognostic effects of GM tests obtained at various time points and sampling sites in a hospital-based, real-world study.

Our study has some limitations. First, due to the retrospective study design, missing data could not be avoided. Bronchoscopy was not done to obtain a GM test of BAL samples in every participant. However, our study reflected the pattern of clinical practice for the diagnostic approach and management of IPA in a hospital-based setting. Second, our study enrolled participants from a single medical center in Taiwan. Our results cannot be extrapolated to other hospitals or patient groups of different ethnicities. Nonetheless, as our study was conducted in a single institute, the homogeneity of treatment flow for IPA minimized the possibility of physicians’ behavior-related bias. Third, the case numbers of different IPA groups were small, and therefore the statistical analyses might have been underpowered for the detection of significant variables contributing to deleterious outcomes. In addition, we did not include a subgroup of influenza-associated pulmonary aspergillosis (IAPA) or coronavirus disease 2019 associated pulmonary aspergillosis (CAPA) in the analysis. IAPA is increasingly recognized as having specific features, but shows a poor correlation with classical risk factors, lacks typical radiological features, and has different case definitions (Schauwvlieghe et al., 2018; Verweij et al., 2020). CAPA may be linked to the rising utilization of dexamethasone and anti-interleukin-6 therapies such as tocilizumab (Bartoletti et al., 2021; Koehler et al., 2021). Further prospective multicenter studies are warranted.

Conclusion

Our study demonstrated that higher ODI of GM test in either serum or BAL fluid samples was related to dismal outcomes. We identified comorbidities associated with respiratory failure and renal damage. Physicians should be aware of the high risk of putative IPA, even in the absence of traditional risk factors, and should provide timely intensive care for this group of patients.

Supplemental Information

Supplemental Information 1 Raw data for tables and figures.

List of abbreviations

IPA invasive pulmonary aspergillosis

GM galactomannan

ODI optical density index

CRP C-reactive protein

CKD chronic kidney disease

DM Diabetes mellitus

COPD chronic obstructive pulmonary disease

EORTC/MSGERC European Organization for Research and Treatment of Cancer/Mycosis Study Group Education and Research Consortium

BAL bronchial alveolar lavage

WBC white blood cell count

HR hazard ratio

CI confidence interval

IFD invasive fungal disease

IAPA influenza associated pulmonary aspergillosis

CAPA coronavirus disease 2019 associated pulmonary aspergillosis.

Additional Information and Declarations

Competing Interests

Author Contributions

Human Ethics

Data Availability

The authors declare that they have no competing interests.

Wei-Che Chen conceived and designed the experiments, prepared figures and/or tables, and approved the final draft.

I-Chieh Chen performed the experiments, analyzed the data, authored or reviewed drafts of the article, and approved the final draft.

Jun-Peng Chen performed the experiments, analyzed the data, prepared figures and/or tables, and approved the final draft.

Tsai-Ling Liao performed the experiments, prepared figures and/or tables, and approved the final draft.

Yi-Ming Chen conceived and designed the experiments, authored or reviewed drafts of the article, and approved the final draft.

The following information was supplied relating to ethical approvals (i.e., approving body and any reference numbers):

The Ethics Committee of Clinical Research, Taichung Veterans General Hospital approved the study (CE21478A).

The following information was supplied regarding data availability:

The raw data for all analysis are available in the Supplemental File.

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
