# Peer review of "Prognostic factors and outcomes of invasive pulmonary aspergillosis, a retrospective hospital-based study"

_PeerJ, doi:10.7717/peerj.17066_

## Round 0.1 · original submission · Minor Revisions

Dear Dr. Chen, the reviewers agree to accept your manuscript.
However, you should revise your paper following the minor comments raised by some reviewers.

Reviewer 1 ·

Basic reporting

No comment

Experimental design

no comment

Validity of the findings

no comment

Reviewer 2 ·

Basic reporting

The justification in the introduction is adequate and appropriate for the study.
The literature review and limitations were mentioned appropriately.

Experimental design

The study methodology and statistics were appropriate

Validity of the findings

The authors aimed to identify the risk factors associated with dismal outcomes in each category of IPA. This study highlighted the importance of high ODI of GM test and timely intensive care of putative IPA patients to minimize mortality.
This information helps to direct future studies to validate these findings.

Additional comments

EORTC-MSGERC criteria in IPA diagnosis is intended to harmonize the clinical trials to understand their results better. I congratulate the authors for their efforts.

Reviewer 3 ·

Basic reporting

no comment

Experimental design

- In line 176, please indicate if two-sided or one-sided p-values were used.
- In lines 185-188, please include the p-values at the end of each sentence to support each statement.
- Any missing data from the covariates? Any missing data in Figure 1, and tables 2-4? Could you include the number of samples included in each logistic regression in the tables?

Validity of the findings

no comment

Reviewer 4 ·

Basic reporting

English is proper. References are appropriately selected and up to date. Raw data are available.

Experimental design

The methodology of the study involves a retrospective analysis of patients diagnosed with invasive pulmonary aspergillosis (IPA) at Taichung Veterans General Hospital in Taiwan from September 2002 to May 2021. The inclusion criteria for possible, probable, and proven IPA were based on the EORTC/MSGERC criteria, and additional patients were classified as putative IPA using the AspICU algorithm. The study was approved by the Ethics Committee of Clinical Research. Data were collected from electronic health records, including demographics, laboratory profiles, comorbidities, and outcomes. Galactomannan antigen tests were performed, and outcomes such as respiratory failure, kidney failure, and mortality were assessed. The study provides a comprehensive evaluation of IPA in the specified population.

I suggest adding in the text the Ethics Committee agreement number.

Validity of the findings

The study identified several risk factors for respiratory failure, kidney failure, and mortality in patients with invasive pulmonary aspergillosis (IPA). Independent risks for respiratory failure included higher first GM test ODI in BAL fluid samples, liver cirrhosis, and solid organ malignancy. Patients classified as probable IPA had a higher risk for respiratory failure compared to those classified as possible IPA. For kidney failure, independent risks included higher serum creatinine value, higher GM test ODI in serum or BAL fluid samples, diabetes mellitus, and post-liver or kidney transplantation. Hemoglobin levels and hematological disease were associated with a lower risk of renal failure. Mortality risks were associated with higher CRP levels and GM test ODIs. Putative IPA had a higher risk of mortality compared to possible IPA. The 3-year survival rates varied among IPA groups, with putative IPA having lower rates compared to the other IPA groups. The findings highlight the importance of considering these factors in the management of patients with IPA.

Additional comments

I suggest adding in the text the Ethics Committee agreement number.

---

## Round 0.2 · accepted · Accept

The reviewers have shown positive feedback after the revision of your manuscript.

Reviewer 3 ·

Basic reporting

The authors have adequately addressed my comments. Therefore, I have no further comments.

Experimental design

The authors have adequately addressed my comments. Therefore, I have no further comments.

Validity of the findings

The authors have adequately addressed my comments. Therefore, I have no further comments.

Reviewer 4 ·

Basic reporting

The authors corrected the manuscript according to the reviewer's suggestions. I recommend the article for publication.

Experimental design

The authors corrected the manuscript according to the reviewer's suggestions. I recommend the article for publication.

Validity of the findings

The authors corrected the manuscript according to the reviewer's suggestions. I recommend the article for publication.